# Physiological and Psychological Effects of Volatile Organic Compounds from Dried Common Rush (*Juncus effusus* L. var. *decipiens* Buchen.) on Humans

**DOI:** 10.3390/ijerph19031856

**Published:** 2022-02-07

**Authors:** Minkai Sun, Taisuke Nakashima, Yuri Yoshimura, Akiyoshi Honden, Toshinori Nakagawa, Yu Nakashima, Makoto Kawaguchi, Yukimitsu Takamori, Yoshitaka Koshi, Rimpei Sawada, Shinsuke Nishida, Koichiro Ohnuki, Kuniyoshi Shimizu

**Affiliations:** 1School of Architecture and Urban Planning, Suzhou University of Science and Technology, Suzhou 215000, China; 2713@usts.edu.cn; 2Department of Agro-Environmental Sciences, Faculty of Agriculture, Kyushu University, Fukuoka 8190395, Japan; nakashima.taisuke.728@m.kyushu-u.ac.jp (T.N.); yuriyoshimura530@gmail.com (Y.Y.); mmcrc764@yahoo.co.jp (A.H.); nakagawa.t@ses.usp.ac.jp (T.N.); 3Department of Biological Resources Management, The University of Shiga Prefecture, Hikone 5228533, Japan; 4Kumamoto Prefectural Agricultural Research Center Agricultural System Research Institute, Yatsushiro 8694201, Japan; nakashima-y-dd@pref.kumamoto.lg.jp (Y.N.); kawaguchi-m-dv@pref.kumamoto.lg.jp (M.K.); ytakamori@gmail.com (Y.T.); qys02755@nifty.com (Y.K.); sawada-r-dh@pref.kumamoto.lg.jp (R.S.); nishida-s-dh@pref.kumamoto.lg.jp (S.N.); 5Faculty of Humanity-Oriented Science and Engineering, Kindai University, Iizuka 8200011, Japan; ohnuki@fuk.kindai.ac.jp

**Keywords:** common rush (*Juncus effusus* L. var. *decipiens* Buchen.), heart rate variability (HRV), alpha component, subjective evaluation, volatile organic compounds (VOCs), electroencephalogram (EEG)

## Abstract

This study compared the participants’ physiological responses and subjective evaluations of air scented with different concentrations of common rush (*Juncus effusus* L. var. *decipiens* Buchen.) (30 g and 15 g, with fresh air as a control). We asked 20 participants to complete a series of visual discrimination tasks while inhaling two different air samples. We evaluated (1) brain activity, (2) autonomic nervous activity, and (3) blood pressure and pulse rate, (4) in combination with self-evaluation. In addition, we quantified the concentrations of volatile organic compounds. The participants reported the scent to be sour, pungent, and smelly; this impression was likely caused by hexanal and acetic acid. Although the self-evaluations showed that participants did not enjoy the scent, their alpha amplitudes of electroencephalogram and parasympathetic nervous activity were increased, suggesting that participants were relaxed in this atmosphere. Moreover, a lower concentration resulted in a greater induction of relaxation. While the air was not pleasant-smelling, the volatile organic compounds present had a positive psychophysiological impact.

## 1. Introduction

The quality of interior space is affected by many factors, such as temperature, humidity, and visual design [1,2,3,4,5]. Smells can influence people’s feelings about a room by changing their physiological and psychological responses [6,7,8]. Furthermore, the inhalation of volatile organic compounds (VOCs) emitted from interior materials is related to health conditions [9,10,11]. Natural materials, in particular, have been considered to enhance physiological well-being [12,13,14].

Recent studies focused on natural materials to investigate whether VOCs from natural, fresh, dried, or produced plants affect humans’ physical and psychological responses through inhalation. For instance, Yatagai demonstrated that individuals who inhaled the scent of Japanese cedar (*Cryptomeria japonica*) reported a “natural” and “peaceful” feeling; their pulse rates were slightly decreased [15,16]. The ratio of high to low frequency (HF/LF) of an electrocardiogram (heart rate variability [HRV]) and salivary alpha-amylase levels of participants increased during and after arithmetic work in a room scented with *C*. *japonica*, which suggested that the scent suppressed stress increments [10]. In another study, the hemoglobin concentration in cerebral blood flow decreased along with blood pressure when people were exposed to the scent of wood [9]. Participants who inhaled VOCs from *Vetiveria zizanioides* had faster reaction times to visual tasks and maintained high sympathetic nervous system activity after a visual discrimination task [17]. After inhaling the smell of pine needles, participants’ cerebral activity was enhanced in feeling-, judgment-, and motor-related brain areas in the frontal lobe and memory areas in the temporal lobe. Self-evaluation showed that the scent seemed natural [14]. After inhaling VOCs from *Acer truncatum* Bunge and *Cedrus deodara*, participants’ blood oxygen saturation increased and their blood pressure decreased; 50% of the participants considered the scent of both objects as “natural” [18].

Common rush (*Juncus effusus* L. var. *decipiens* Buchen.) is a perennial herbaceous plant that has been widely cultivated in Japan since ancient times. Dried stalks of common rush are used to produce the covers of tatami mats, which are widely used as floor covering. Tatami mats induce feelings of calmness and coziness [19]. The present study showed that participants preferred a room with tatami; this preference can be partially ascribed to culture and traditional customs.

A previous study established an experimental method for evaluating the effect of VOCs based on changes in physiological and psychological responses when performing a visual discrimination task [20]. The study also showed that visual discrimination tasks forced participants to concentrate; therefore, it can be assumed that the two groups of participants in that study focused on the same stimuli.

To further investigate how VOCs influence participants’ brain activity, we employed an electroencephalogram (EEG). A previous study showed that the EEG is a suitable indicator for assessing the effects of VOCs from natural materials [21]. In particular, alpha activity (7.5–12.5 Hz) is the dominant component of amplitude during relaxed and eyes-closed wakefulness [22]. While theta (4–7 Hz) frequencies are usually prominent during drowsiness [23], theta band activities orchestrate memory reactivation during wakefulness and sleep, and play an essential role in memory consolidation [24,25].

Therefore, in order to clarify how VOCs from common rush affect the physiological and psychological responses of humans, the present study recorded the alpha and theta frequencies of the EEG during visual tasks. Moreover, HRV and self-evaluation were conducted to reinforce reliability while revealing the relationship between VOCs and responses. The results in combination with the physiological and psychological responses to common rush are discussed here.

## 2. Materials and Methods

### 2.1. Participants

Twenty healthy students (12 females and 8 males; aged 18–28 years) at Kyushu University were recruited as participants in this study. They were asked to avoid drinks containing caffeine, smoking, any kind of drug, and vigorous physical activity, and to have a restful night’s sleep before the experiments. The Human Research Ethics Committee of Kyushu University approved the experimental protocol. The participants received 10,000 JPY (approximately 95 USD) as compensation. This study was conducted in accordance with the Declaration of Helsinki.

### 2.2. Experimental Room

Figure 1 shows the experimental room and the apparatus used in the experiment. The air from the pressure gas cylinder was delivered 15 cm in front of the participant for 20 min; the air passed through a bottle containing activated carbon, an empty bottle, a flow meter, and a chamber containing the sample, before getting delivered. The experimental apparatus was controlled via a constant-flow olfactometer (CMQ0005B; Yamatake Co., Yamaguchi, Japan) that kept the air at a constant rate of 1.0 L/min through Teflon tubing in a charcoal trap, and then into an empty trap to eliminate any charcoal particles. The air was made to flow into a 500-mL plastic chamber with or without the dried common rush, which was cut into 1.5 cm. Air containing VOCs from common rush flowed through a stainless-steel pipe into an industrial acoustics sound room. The indoor temperature of the room (inner size: 1265 × 1760 × 1964 mm, AVITECS MYROOM II; Yamaha Corp., Shizuoka, Japan) was maintained at 25 °C during all of the experiments. The participants sat on a chair made of stainless steel pipes, which were placed in front of a funnel that delivered the airflow. The airflow exited a funnel fixed at a height of 1 m. The pipes, tubes, funnel, and traps were washed with acetone after each sample was delivered. A vent was used to eliminate any residual VOCs in the room (Figure 1).

### 2.3. Sample Determination

A previous study showed that different concentrations of VOCs have distinct effects on individuals who inhale them [26]. Thus, a preliminary experiment was conducted to determine a suitable quantity for the high and low doses of common rush. Six samples of 1.5 cm of dried common rush were used at different weights (1.875, 3.75, 7.5, 15, 30, and 60 g). Twelve participants were recruited as volunteers and were asked to smell and evaluate the concentrations at five levels (Figure 2). The low and high quantities were designated as “threshold (slight smell)” and “strong (strong smell),” respectively. The two samples of common rush that were most frequently statistically evaluated (Statistical Package for Social Sciences [SPSS], mode value) were defined as “threshold” at 15 g (low-quantity [LG]) and “strong” at 30 g (high-quantity [HG]) of common rush, with fresh air as the control.

### 2.4. Visual Discrimination Task

We used a visual discrimination task during the inhalation period to (1) keep the participants’ minds focused on the task and (2) simulate an office environment. The visual discrimination task consisted of six sessions. Each session was 2.5 min long, and the content was identical. Between each session, the participants rested for 1 min. The task lasted for approximately 20 min. The participants were tested individually in a room wherein all noise from outside the room was reduced to less than 20 dBA. The light inside the room was then turned off; the only light source was a cathode-ray tube (CRT) monitor. The order of exposure to each sample was counterbalanced across the participants. The participants were seated in front of a 16-inch color CRT monitor, (Image Quest Q770; Hyundai Motor Company, Seoul, South Korea) controlled using Presentation Ver. 16.3 (Neurobehavioral Systems, Inc., San Francisco, CA, USA) on a personal computer (Windows 7 Professional; Microsoft Corporation, Redmond, WA, USA). The viewing distance was 114 cm. The visual discrimination task consisted of five stimulus patterns (vertical, 5% right tilt, 5% left tilt, 10% right tilt, and 10% left tilt) of a figure that appeared on the monitor every 1 s, and each pattern was presented for 200 ms. The participants were asked to click a mouse when a designated pattern appeared. Reaction times and response accuracies during the tasks were recorded, and the average values across the sessions were calculated (Figure 3).

### 2.5. Physiological Responses

The EEG was used as a metric to evaluate the participants’ physiological responses. The EEG waveforms were recorded using active gold plate electrodes that recorded EEGs with high impedance (Polymate V, AP5148; Miyuki Giken Co., Ltd., Tokyo, Japan). The recording electrodes were placed on a flexible cap using the international 10–20 system. All of the recording electrodes referred to electrodes on the ear lobes. The EEG sampling frequency was 1000 Hz. We applied second-order Butterworth filters for low-cut filtering and first-order Butterworth filters for high-cut filtering in the range of 0.5–50 Hz in offline analyses. The electrode gel from Electro-Caps (Electro-Cap International, Inc., Eaton, Ohio, USA) was applied to each active electrode to keep the impedance below 30 kΩ. EEG frequency analyses were conducted using a custom MATLAB script (ver. 9.1; Mathworks, 294 Sherborn, MA, USA).

HRV is a sensitive and noninvasive tool for evaluating the activity of human autonomic nervous system [27]. In this study, the two major spectral components of HRV, the low frequency (LF; 0.04–0.15 Hz) and high frequency (HF; 0.15–0.4 Hz) bands, were calculated using complex demodulation maximum-entropy methods [28]. The LF/HF ratio of the R-R interval variability was also assessed (R-R Interval CDM Analysis; NoruPro Light Systems, Inc., Tokyo, Japan). HF reflects parasympathetic nervous activity, and the LF/HF ratio reflects sympathetic nervous activity [29,30]. HRV was recorded using Polymate V (Miyuki Giken Co., Ltd., Tokyo, Japan) for 20 min during the visual tasks. Two electrodes were attached to each participant’s left and right wrists to record HRV. Blood pressure and pulse rate were also measured after the participants entered and before they exited the room (Terumo Electronic Sphygmomanometer P2000; Terumo Corporation, Tokyo, Japan).

### 2.6. Psychological Responses

A visual analog scale (VAS) was used to identify the position of each word along a continuous line (10 cm). The participants were asked to mark the line that intuitively matched their perceptions regarding a smell. The questionnaire contained seven words related to distinct perceptions: sweetness (0–100), bitterness (0–100), astringency (0–100), sourness (0–100), smell (0–100), fragrance (0–100), and freshness (0–100). To assess the mood, the short form of the Profile of Mood States (POMS-SF) was used [31]. The participants were asked to check a five-point scale ranging from “not at all” to “extremely” for all 30 questions after they exited the room. The POMS-SF questions were assigned to six different dimensions of mood, including tension/anxiety (T/A), anger/hostility (A/H), vigor (V), fatigue (F), depression (D), and confusion (C). Subsequently, T-scores were calculated by following the manual to assess mood states quantitatively [31].

### 2.7. Gas Chromatography-Mass Spectrometry Analysis

VOCs in the air that presented in HG (30 g of common rush) and LG groups (15 g of common rush) were collected in Tedlar bags (20 L).

The air conditioning had the same settings for each experiment. The VOCs in the Tedlar bags were collected in sorbent tubes (Tenax TA; Gerstel GmbH & Co.KG, Mülheim an der Ruhr, Germany) via a pump (flow rate, 0.15 L/min; amount, 3 L; duration, 20 min). The collection was repeated three times for each Tedlar bag, and six tubes were obtained for VOC adsorption. The collected VOCs in each tube were analyzed using a gas chromatograph-mass spectrometer (GC-MS; Agilent 7890A GC/5975C MS, Agilent Technologies, Palo Alto, CA, USA) system with the Thermal Desorption Unit 2 (TDU2; Gerstel GmbH & Co.KG) and Cooled Injection System 4 (CIS4; Gerstel GmbH & Co.KG). 

The initial temperature of desorption was maintained at 40 °C for 0.1 min; the temperature was then raised to 220 °C at 720 °C/min and held for 3 min. The desorbed compounds were then cryo-focused on the CIS4, cooled to −100 °C, and held for 0.1 min with liquid nitrogen. At the end of desorption, the CIS4 temperature was raised to 220 °C at 12 °C/min and held for 10 min to inject the compounds into the column. The GC-MS was equipped with a VF-624 column (60 m × 0.25 mm; film thickness, 1.4 μm; Agilent Technologies, Palo Alto, CA, USA). The oven temperature was raised from 60 to 280 °C at a rate of 10 °C/min and held at this temperature for 8 min. Helium was used as the carrier gas at a flow rate of 1 mL/min. The split ratio was set to 20:1.

For the quantitative analysis of n-hexanal, 1 µL of benzaldehyde diluted with acetone (standard quantity, 200 µL/L) was added to each absorption tube as the internal standard. The quantification of n-hexanal was performed using a calibration curve based on peak area data from selected ion monitoring, relative to the internal standard.

VOCs were identified by comparing the mass spectra with that of the U.S. National Institute of Standards and Technology (NIST) mass spectral library (11.0), Aroma office (Nishikawa Keisoku Co., Ltd., Tokyo, Japan), and retention index.

### 2.8. Experimental Setting

Central air conditioning was used to set the temperature and humidity within the respective ranges of 25–27 °C and 40–50%. On the day after the participants arrived, they were asked if their physical condition was good. Moreover, to prevent the participants from leaving the room during the experiment, they were asked to use the restroom before being led to the experimental chamber. Once in the room, they were asked to sit on a chair for 5 min to calm down. After they had calmed down, their blood pressure and pulse were measured. The participants were instructed to fill out the POMS-SF form. After the EEG device was set up, the participants were asked to enter the experimental chamber and sit on a chair for the visual discrimination task. After the visual discrimination task, blood pressure and pulse were measured again. They were then asked to fill out the second POMS-SF form, followed by a VAS questionnaire. All of the experiments lasted approximately 2 h (Table 1, Figure 4). To avoid sequence effects, the order of exposure was determined randomly. To avoid carry-over effects of the samples, a 1-w interval was set. All of the experiments were performed between October and November 2017.

### 2.9. Data Analysis

The alpha components of the EEG were calculated using custom MATLAB scripts. HRV was analyzed using R-R Interval (CDM) analysis software (NoruPro Light Systems, Tokyo, Japan). The T-scores of POMS-SF were calculated according to a manual [31]. The averages of the blood pressure and pulse rates recorded after entering and before exiting the chamber were calculated. The HF and LF/HF scores were changed in the natural log to prevent individual differences. Repeated-measures analyses of variance with multiple comparisons corrected by Bonferroni methods were conducted on EEG, HRV, mean blood pressure, mean pulse, POMS-SF scores, VAS questionnaires, reaction times, and accuracies during the task. In all cases, the significance level was set at *p* < 0.05, and the marginal significance was set at *p* < 0.1. The statistical analyses were performed using SPSS version 19.0 software (IBM Corporation, Armonk, NY, USA).

## 3. Results

### 3.1. Physiological Responses

The present study analyzed brain wave data from six channels: Oz, O1, and O2 from the occipital lobe, and Fz, F3, and F4 from the frontal lobe. However, the main effects were not observed in the occipital or frontal lobe. The results of multiple comparisons showed that the alpha amplitudes at Oz were significantly increased compared to that at fresh air when the participants inhaled the LG sample (0.058 ± 0.028 µV vs. 0.102 ± 0.038 µV, *p* < 0.05) (Figure 5).

Further studies should test whether these VOCs could be harmful in the long term.

Figure 6 shows the natural logarithms of HF (ln HF), which reflect the parasympathetic nervous activity and values of ln (LF/HF), estimating the participants’ sympathetic nervous activity while performing tasks. There was a significant main effect of the samples (F [2, 36] = 4.93, *p* < 0.05, ηp2 = 0.22). There was a significant and marginal difference in the HF component between the HG (2.95 ± 0.10 vs. 3.12 ± 0.81, *p* < 0.05), LG (2.95 ± 0.10 vs. 3.13 ± 0.75, *p* < 0.1) (Figure 6), and C samples. Blood pressure and pulse rate showed no significant differences between the samples.

### 3.2. Psychological Responses

The results of the VAS scores showed that there were significant main effects of the samples (F [2, 36] = 9.35, *p* < 0.01, ηp2 = 0.33) in terms of astringency. Compared to fresh air (13.35 ± 3.17), the participants perceived significant astringency in both HG (35.55 ± 6.4, *p* < 0.01) and LG (32.40 ± 4.42, *p* < 0.01) samples. There was a significant main effect of the samples (F [2, 36] = 8.51, *p* < 0.01, ηp2 = 0.31) in terms of sourness. Compared to fresh air (11.85 ± 3.21), the participants perceived that both the HG (34.25 ± 7.06, *p* < 0.01) and LG (37.45 ± 6.81, *p* < 0.05) samples were significantly sour. There was a significant main effect of the samples (F [2, 36] = 13.97, *p* < 0.01, ηp2 = 0.42) in terms of smelliness. Compared to fresh air (15.90 ± 5.33), the participants perceived both HG (48.90 ± 7.17, *p* < 0.01) and LG (47.30 ± 5.12, *p* < 0.01) samples as being significantly smelly (Figure 7). No significant differences were found in the POMS-SF scores among the three samples.

### 3.3. Visual Discrimination Task

The participants’ reaction times were less than 550 ms during all six sessions, without significant differences among all three types of samples; the participants’ accuracy did not vary. The participants maintained an accuracy of greater than 85% during all six sessions, without significant differences among all three types of samples.

### 3.4. Volatile Organic Compounds

Acetic acid, pentanal, propanoic acid, hexanal, *cis*-3-hexene-1-ol, 2-ethyl-1-hexanol, and benzaldehyde were detected. In each condition, the concentration of hexanal, the main volatile component of common rush, was 7.6 ± 0.3 ng/L and 23.7 ± 0.9 ng/L in the low- and high-quantity groups, respectively. The high-concentration group had three times the concentration of the low-concentration group. The concentration range of n-hexanal in the soundproof room (4.24 m^3^) where the participants were tested was calculated to be 0.112–23.7 ng/L and 0.036–7.6 ng/L in the high- and low-concentration groups, respectively.

## 4. Discussion

In the present study, the alpha component placed at each participant’s occipital lobe showed significant increases during the inhalation of the LG sample of common rush. Previous studies have shown that the alpha component can increase when participants perceive scents produced by natural plants [32]. One of the compounds, benzaldehyde, was used as a chemical flavoring, particularly to mimic “cool mint”. A previous study suggested that benzaldehyde could have a subjectively refreshing effect in humans [33].

Regarding the alpha component, an increasing alpha component is considered a metric of relaxation and coziness, and the neural origin is derived from the occipital lobe [23,34,35]. Previous studies have reported that when people felt relaxed, calm, cozy, or were in a low-stress condition, their alpha amplitude of EEG increased [23,34,35]. A study found that increase in the alpha and theta powers at the occipital lobes were associated with reduced attention-related arousal [36]. Thus, this result suggests that VOCs from common rush could help people to relax while working in an office. Moreover, frontal alpha oscillations were associated with efficient neurocognitive processing when concentrated on Stroop tasks [37]. Cortical oscillations in the alpha frequency band were correlated with creative thinking, and alpha recruitment increases played an essential role in creative ideation [38,39,40]. These results of previous studies can partially explain why participants maintained their accuracy and reaction time during the visual discrimination task. The present study could not discriminate which compound effectively enhanced the alpha component; therefore, further research should identify the effective compound for alpha activities. The theta component did not change between the three samples, suggesting that the participants felt relaxed but not sleepy.

The results of HF component showed that compared to fresh air, an increase in parasympathetic nervous activity was observed when the participants inhaled VOCs from the common rush. As ln HF is an indicator of parasympathetic nervous activity, the results of the present study imply that autonomic relaxation in the participants was induced during the experiment. These results are in line with those of previous studies, wherein parasympathetic nervous activity increased when participants were surrounded by forest environments or inhaled VOCs from natural plant elements [18]. Combined with an increase in the alpha component, these two pieces of evidence suggest that VOCs from common rush could keep people relaxed during computer tasks without sacrificing reaction times and accuracy.

Self-evaluation revealed that the participants felt that common rush was sour, astringent, and smelly. Basically, dominant compounds, such as hexanal and acetic acid, have unpleasant scents. The VOCs of common rush contain a high percentage of hexanal and acetic acid, eliciting negative evaluations. On the other hand, the results of the POMS-SF showed that the participants were not stressed. They maintained high reaction speed and accuracy during the task. A previous study demonstrated that n-hexanal stimulated dopamine release in rats [41]. Another study found that if dopamine receptors were blocked, the wakefulness of rats decreased [42], suggesting that dopamine can play an important role in wakefulness. Therefore, we infer that n-hexanal from common rush enhanced participants’ dopamine release and maintained participants’ arousal during the task while keeping them relaxed.

Although self-evaluation showed that participants did not like the smell of common rush, the physiological responses clearly showed the benefits of maintaining high task performance and making participants more relaxed as a result of inhaling its VOCs. Otherwise, EEG data showed that it was even more effective when participants inhaled a low quantity of common rush that was barely perceptible. Further studies should focus on eliminating the unpleasant smells of common rush while retaining the effects of its volatile compounds.

## 5. Conclusions

To date, human physiological and psychological responses to VOCs from common rush have rarely been investigated. One of the most significant findings to emerge from this study is that VOCs from common rush could have considerable psychological and emotional effects on people, especially those with a desk job in an office. Although self-evaluation showed that the smell of common rush was not favorable, the increasing alpha component and parasympathetic nervous activity suggested that the VOCs from the plant kept people relaxed when stress was introduced. In addition, as mentioned above, both physiological and psychological responses play a role in human interaction with VOCs. The present study provides a more in-depth insight into the new value and function of common rush, as it supports people’s well-being.

## 6. Limitations

Common rush is a natural material that has been used in Japan since ancient times, and its VOCs are not considered harmful. However, the effects of extremely high concentrations beyond those used in this study remain unclear. Further studies should test this potential. Moreover, the number of participants in the present study was relatively small. Further studies should involve a larger number of participants.

## Figures and Tables

**Figure 1 ijerph-19-01856-f001:**
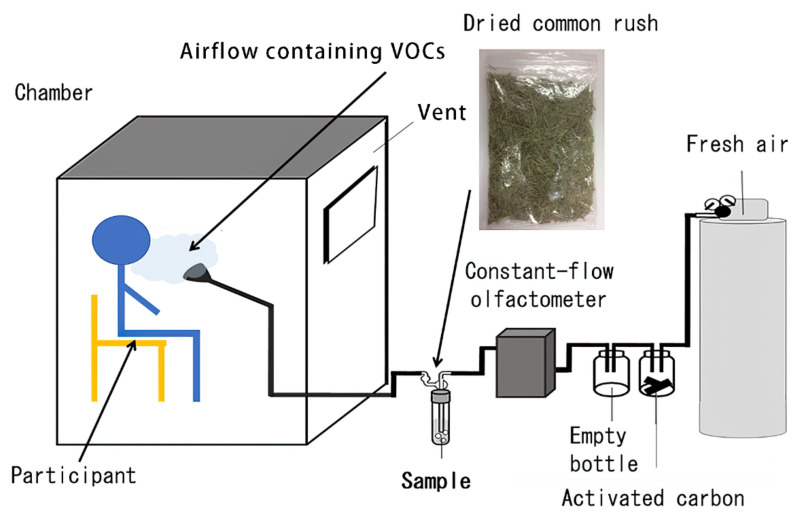
Experiment setting.

**Figure 2 ijerph-19-01856-f002:**
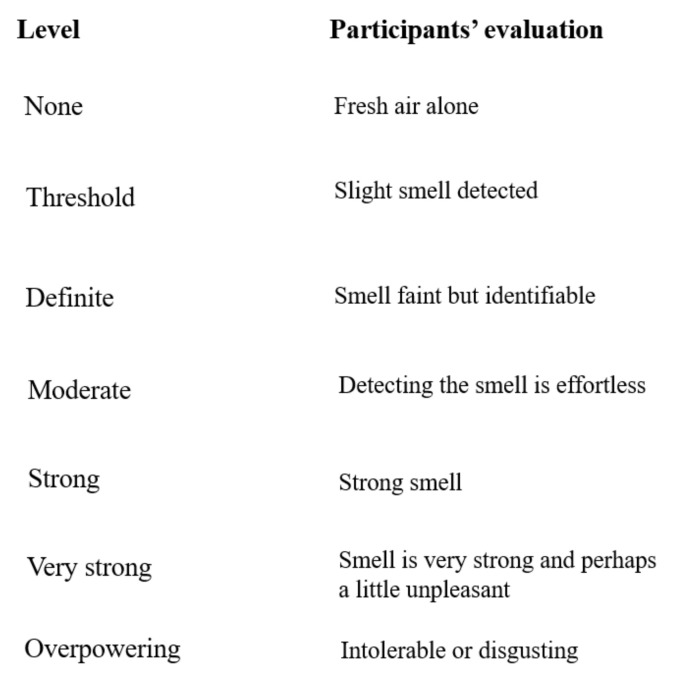
Sheet used in the preliminary experiment (five levels of concentration of the smell).

**Figure 3 ijerph-19-01856-f003:**
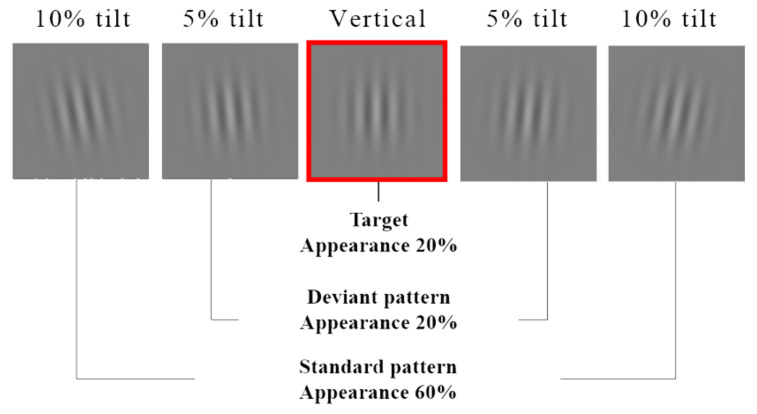
The five stimulus patterns of the figure displayed on the monitor during the visual discrimination task.

**Figure 4 ijerph-19-01856-f004:**
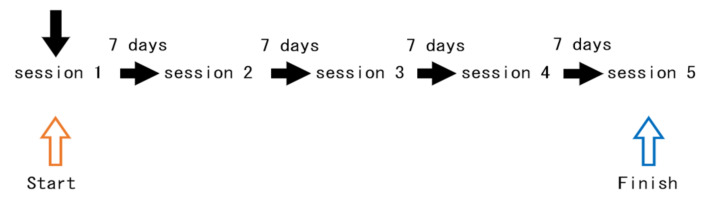
Schedule for the experiment.

**Figure 5 ijerph-19-01856-f005:**
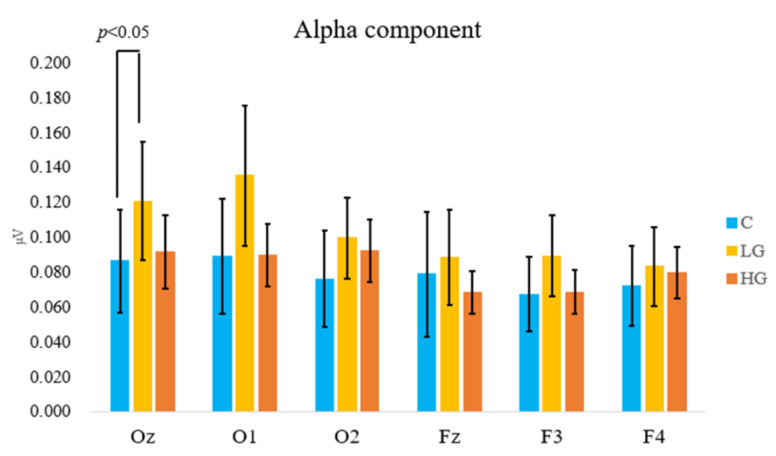
Comparison of the alpha component at Oz in fresh air (C), low-quantity (LG), and high-quantity (HG) samples during the task.

**Figure 6 ijerph-19-01856-f006:**
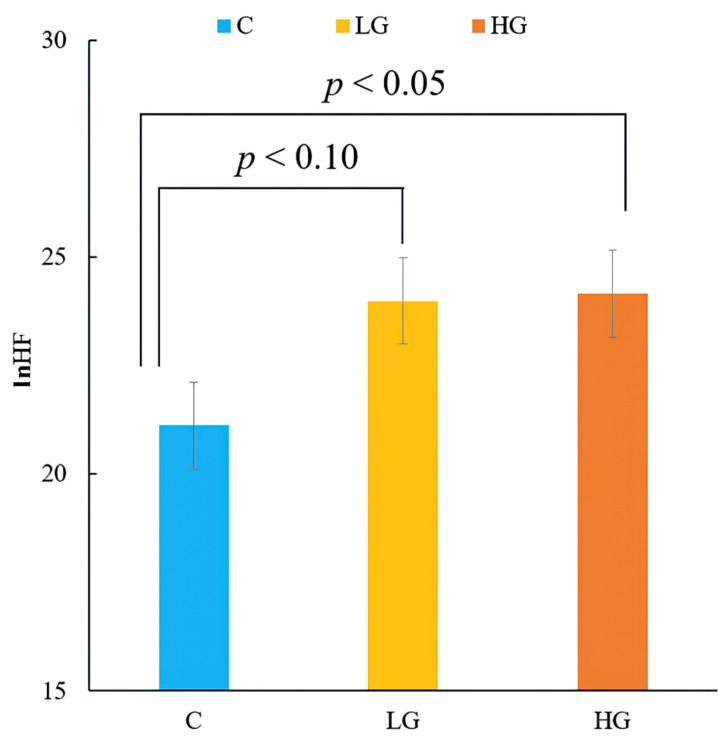
Comparison of ln HF in fresh air (C), low-quantity (LG), and high-quantity (HG) samples.

**Figure 7 ijerph-19-01856-f007:**
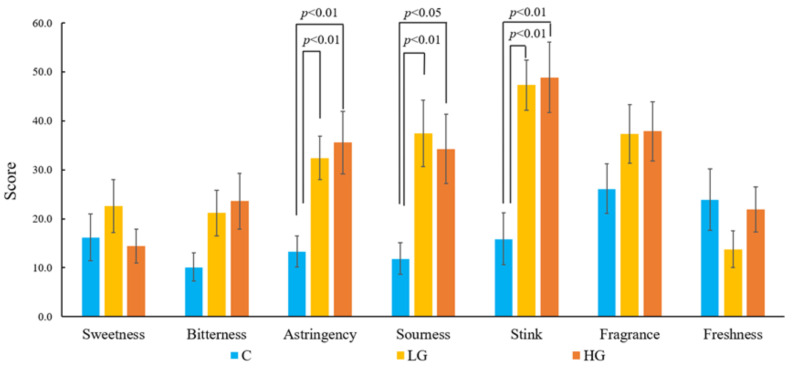
Comparison of the VAS scores of the fresh air (C), low-quantity (LG), and high-quantity (HG) samples during the task.

**Table 1 ijerph-19-01856-t001:** Timetable for one session.

	Preparation (50 min)	Pre-Test (10 min)	Task (20 min)	Post-Test (15 min)	Remove Device (30 min)
Device fixing	●				
Condition check	●				
Blood pressure and pulse		●		●	
POMS		●		●	
EEG and HRV during visual discrimination task			●		
VAS and POMS				●	
Remove device					●

●: conduct.

## Data Availability

Not acceptable.

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
