# Peer review of "Physiological and Psychological Effects of Volatile Organic Compounds from Dried Common Rush (*Juncus effusus* L. var. *decipiens* Buchen.) on Humans"

_ijerph, 2022, doi:10.3390/ijerph19031856_

Round 1
Reviewer 1 Report
In my opinion, the paper is well structured, the research methodology, the discussion and the results are explained in detail. I have some questions and some recommendations.
Firstly, about the questions:
- In Figure 3, I cannot see the differences among the images. Please, could you give more details about these images?
- Your research analyses the effect of the VOCs in people in a short period of time, but these substances could be harmful in the long term. Have you considered the study of the negative effects in the long term?
Secondly, as regards the recommendations,
- You should revise the English language throughout the paper.
- You should explain the meaning of letters C, LG and HG in Figures 5 and 6.
Author Response
In my opinion, the paper is well structured, the research methodology, the discussion and the results are explained in detail. I have some questions and some recommendations.
Firstly, about the questions:
- In Figure 3, I cannot see the differences among the images. Please, could you give more details about these images?
■Thank you for your comment. We changed the figure to make the images more distinguishable. We also added “(vertical, 5% right tilt, 5% left tilt, 10% right tilt, and 10% left tilt)” on page 4, lines 137-138 to explain the differences among the five images.
- Your research analyses the effect of the VOCs in people in a short period of time, but these substances could be harmful in the long term. Have you considered the study of the negative effects in the long term?
■Thank you. Some substances may have an unpredicted influence on humans in the long term. We did not test it in the present study. We add, “Although common rush is a natural material, some of its VOCs can be harmful in the long term. Further studies should test this potential. Moreover, the number of participants in the present study was relatively small. Further studies should involve a larger number of participants.” in the limitation part. For further study, we will test if there were any negative effects of substances.
- You should revise the English language throughout the paper.
■Thank you. A native English speaker revised the language throughout the paper.
You should explain the meaning of letters C, LG, and HG in Figures 5 and 6.
■Thank you. We add an explanation of C, LG, and HG in Figures 5 and 6.
■We found several mistakes in the previous manuscript as follows. We have correctly revised them.
- We replaced “Bunchen.” with “Buchen.” in the scientific name of common rush and corrected the Italic and Roman fonts.
- We inserted new 7th author, Makoto Kawaguchi, who belongs to Institute #4
- We inserted the email address of the Co-first author.
- We replaced “subjects” with “participants” in the Informed Consent Statement.
- We changed the telephone number of the correspondent author.
- We replaced “Igusa” with “common rush.”
- We inserted “quantity” to “LG” and “HG” for clarity.

Reviewer 2 Report
Review of paper entitled “Physiological and Psychological effects of volatile organic compounds (VOCs) from dried common rush (Juncus effusus L. var. decipiens Bunchen.) on human”
Dear authors, thank you for submitting this interesting study on the psychophysiological effects of the smell of dried common rush on humans.
The study involved 20 participants that were asked to perform a visual discrimination task while being exposed to VOC from dried common rush in two different concentrations, and a control condition. The results indicate that exposure to VOC had a positive effect on relaxation while performing the task. Interestingly, the positive effect on physiology was not mirrored in the subjective evaluations. The paper would make a welcome contribution to the field, please consider the following minor issues.
Abstract
Good and informative
20 “high and low concentration of 30g and 15g, respectively”. It is difficult to see at this point in the manuscript how g. relates to concentration.
Introduction
45 Not sure, perhaps HRV would be more clear than ECG in this context.
75-77 Is this the aim of the study? I recommend giving some reconsideration of how the aim is formulated. See if it is possible to clarify why different kinds of data were collected (in addition to EEG).
Materials and methods
Please ensure that the control condition is described, perhaps also in abstract.
117 Figure 2 could be smaller. Optional
126 “The participants were tested individually in a room in which all ambient noise and light were cut.” Please clarify; was the room completely dark? Usually, it is not possible to remove all ambient noise (ventilation etc), please clarify what noises were removed. (Rooms perceived as quiet still tend to have about 20-30 dBA of ambient noise.)
147 “filtered in the range of 0.5-50Hz” What kind of filter?
174 I would avoid abbreviations in the title. Also, I can’t find an explanation of this abbreviation, even at first mention.
229 The name of the analysis software appears to be missing.
Results
Figures: Please ensure abbreviations are explained in the figure captions. For instance, figure 5 could be “Comparison of Alpha component at Oz among fresh air (C), low (LG) and high (HG) quantity sample during the task.”
Discussion
Please also mention any limitations with the study, such as the limited number of participants.
336 “and maintain the participants' arousal during the task.” Please elaborate about this statement. Is it possible to be both more relaxed and maintain “arousal”. Perhaps it is different kind of arousal compared to the one experienced during stress. This could be clarified.
340-342 This sentence is a bit unclear. Please consider it.
Conclusion
Good. Thank you for an interesting paper.
Author contributions:
357/359/360 Please check the formulation of manuscript text contribution in line 357 as it appears it is not compatible with lines 359/360.
References:
No reference for number 6.
Author Response
Review of paper entitled “Physiological and Psychological effects of volatile organic compounds (VOCs) from dried common rush (Juncus effusus L. var. decipiens Bunchen.) on human”
Dear authors, thank you for submitting this interesting study on the psychophysiological effects of the smell of dried common rush on humans.
The study involved 20 participants that were asked to perform a visual discrimination task while being exposed to VOC from dried common rush in two different concentrations, and a control condition. The results indicate that exposure to VOC had a positive effect on relaxation while performing the task. Interestingly, the positive effect on physiology was not mirrored in the subjective evaluations. The paper would make a welcome contribution to the field, please consider the following minor issues.
Abstract
Good and informative
- “high and low concentration of 30g and 15g, respectively”. It is difficult to see at this point in the manuscript how g. relates to concentration.
■Thank you for your comments. We deleted the “high and low concentration” and replaced “concentration” with “amount” to avoid unclear descriptions. Similarly, in the results section, we replaced “g” with “ng/L” in P.2 Line 20.
Introduction
- 45 Not sure, perhaps HRV would be more clear than ECG in this context.
■We changed ECG to HRV.
- 75-77 Is this the aim of the study? I recommend giving some reconsideration of how the aim is formulated. See if it is possible to clarify why different kinds of data were collected (in addition to EEG).
■We changed this part to “Therefore, in order to clarify how VOCs from common rush affect the physiological and psychological responses of humans, the present study recorded the alpha and theta frequencies of EEG during visual tasks. Moreover, HRV and self-evaluation were con-ducted to reinforce reliability while revealing their relationship. The results are discussed in combination with the physiological and psychological responses common rush”
Materials and methods
- Please ensure that the control condition is described, perhaps also in abstract.
■We added, “fresh air as a control” in this part and the abstract.
- 117 Figure 2 could be smaller. Optional
■We changed its size.
- 126 “The participants were tested individually in a room in which all ambient noise and light were cut.” Please clarify; was the room completely dark? Usually, it is not possible to remove all ambient noise (ventilation, etc.), please clarify what noises were removed. (Rooms perceived as quiet still tend to have about 20-30 dBA of ambient noise.)
■We added, “The participants were tested individually in a room in which all noise from outside the room was reduced to less than 20 dBA. Then, the light inside the room was turned off; the only light source was a CRT monitor.”
- 147 “filtered in the range of 0.5-50Hz” What kind of filter?
■We applied second-order Butterworth filters for low-cut filtering and first-order Butterworth filters for high-cut filtering. (P5, Line 153-155)
- 174 I would avoid abbreviations in the title. Also, I can’t find an explanation of this abbreviation, even at first mention.
■Thank you, we changed it to “Gas chromatography-mass spectrometry analysis.”
- 229 The name of the analysis software appears to be missing.
■EEG frequency analyses were conducted using custom MATLAB scripts (ver. 9.1, Mathworks, 294 Sherborn, MA). HRV was analyzed using R-R Interval CDM Analysis, NoruPro Light Systems, Inc., Tokyo, Japan. (P.5 Lines163-169)
Results
- Figures: Please ensure abbreviations are explained in the figure captions. For instance, figure 5 could be “Comparison of Alpha component at Oz among fresh air (C), low (LG) and high (HG) quantity sample during the task.”
■We added all the explanations to fig. 5 and fig. 6.
Discussion
- Please also mention any limitations with the study, such as the limited number of participants.
■We added. “Although common rush is a natural material, some of its VOCs can be harmful in the long term. Further studies should test this potential. Moreover, the number of participants in the present study was relatively small. Further studies should involve a larger number of participants. “
- 336 “and maintain the participants' arousal during the task.” Please elaborate about this statement. Is it possible to be both more relaxed and maintain “arousal”. Perhaps it is different kind of arousal compared to the one experienced during stress. This could be clarified.
■We changed this to “Therefore, we infer that n-hexanal from common rush enhanced participants’ dopamine release and maintained participants’ arousal during the task while keeping them relaxed.”
- 340-342 This sentence is a bit unclear. Please consider it.
■We changed it to “Otherwise, EEG data showed that it was even more effective when participants inhaled a low quantity of common rush that was barely perceptible..”
Conclusion
- Good. Thank you for an interesting paper.
Author contributions:
- 357/359/360 Please check the formulation of manuscript text contribution in line 357 as it appears it is not compatible with lines 359/360.
■ We corrected it.
References:
- No reference for number 6.
■ We added reference 6.
■ We found several mistakes in the previous manuscript as follows. We have correctly revised them.
- We replaced “Bunchen.” with “Buchen.” in the scientific name of common rush and corrected the Italic and Roman fonts.
- We inserted new 7th author, Makoto Kawaguchi, who belongs to Institute #4
- We inserted the email address of the Co-first author.
- We replaced “subjects” with “participants” in the Informed Consent Statement.
- We changed the telephone number of the correspondent author.
- We replaced “Igusa” with “common rush.”
- We inserted “quantity” to “LG” and “HG” for clarity.
